# Patient Education and Communication in Palliative Radiotherapy: A Narrative Review

**DOI:** 10.3390/cancers17193109

**Published:** 2025-09-24

**Authors:** Erika Galietta, Costanza M. Donati, Filippo Mammini, Arina A. Zamfir, Alberto Bazzocchi, Rebecca Sassi, Renée Hovenier, Clemens Bos, Milly Buwenge, Silvia Cammelli, Helena M. Verkooijen, Alessio G. Morganti

**Affiliations:** 1Department of Medical and Surgical Sciences, Alma Mater Studiorum University of Bologna, 40138 Bologna, Italy; erika.galietta2@unibo.it (E.G.); costanzamaria.donati@unibo.it (C.M.D.); filippo.mammini@studio.unibo.it (F.M.); milly.buwenge2@unibo.it (M.B.); silvia.cammelli2@unibo.it (S.C.); alessio.morganti2@unibo.it (A.G.M.); 2Radiation Oncology, IRCCS Azienda Ospedaliero-Universitaria di Bologna, 40138 Bologna, Italy; 3Diagnostic and Interventional Radiology, IRCCS Istituto Ortopedico Rizzoli, 40136 Bologna, Italy; alberto.bazzocchi@ior.it (A.B.); rebecca.sassi@ior.it (R.S.); 4Division of Imaging and Oncology, University Medical Center Utrecht, Heidelberglaan 100, 3584 CX Utrecht, The Netherlands; r.hovenier-2@umcutrecht.nl (R.H.); c.bos@umcutrecht.nl (C.B.); h.m.verkooijen@umcutrecht.nl (H.M.V.); 5Division Imaging, Antoni van Leeuwenhoek, Plesmanlaan 121, 1066 CX Amsterdam, The Netherlands

**Keywords:** palliative radiotherapy, patient education, decision aids, pain education, symptom management, bone metastases, advanced cancer, shared decision-making, communication, narrative review

## Abstract

Palliative radiotherapy (PRT) is used to relieve symptoms and preserve quality of life for people with advanced cancer, yet it is often offered late and is frequently misunderstood by patients and clinicians. We reviewed research on structured patient education and on how education and communication are delivered to adults referred to or receiving PRT. Six studies from 2005 to 2023 met our criteria, including two randomized trials. Education delivered at the time of referral or consultation improved knowledge, reduced decisional uncertainty, and increased readiness to proceed with PRT. Education delivered during treatment improved symptom outcomes, with one multicenter trial showing faster and more frequent pain control when pain education accompanied PRT for painful bone metastases. Observational and qualitative studies described limited patient question-asking and persistent curative expectations. Evidence remains scarce and heterogeneous, and pragmatic, scalable education that combines pre-consultation priming, clear goals-of-care language, and post-treatment reinforcement warrants testing.

## 1. Introduction

Palliative radiotherapy (PRT) uses targeted radiation with palliative intent to relieve symptoms, most commonly pain, bleeding, obstruction, or neurologic compromise, and to preserve or improve quality of life in advanced cancer. Compared with curative courses, PRT is typically delivered in short schedules, is generally well tolerated, and can provide rapid and meaningful symptom relief across tumor types and anatomical sites. Accordingly, PRT is considered a cornerstone of modern palliative oncology care [1]. Because the aim of PRT is symptomatic improvement, side effects should be minimized, and any potential toxicities must be carefully balanced against the expected palliative benefit and the patient’s goals of care [1].

Despite its benefits, real-world delivery of PRT is often suboptimal. Population-based data show variability in access and timing, with disparities by age, comorbidity, and race; for example, Black patients with metastatic malignancies have been reported to be less likely to receive PRT in SEER-Medicare cohorts. Moreover, a relevant proportion of patients die shortly after completing PRT, in the order of weeks, suggesting that late referrals limit the chance to realize symptom benefit [2]. Similar concerns emerge from end-of-life cohorts and critical reviews, where delayed initiation curtails the likelihood and magnitude of palliation [3]. When PRT is started very late, symptom gains are necessarily limited: among patients treated in the last two weeks of life, only 26% experienced symptom improvement [4].

Knowledge gaps and misaligned expectations are recurrent barriers to informed, goal-concordant decision-making, timely initiation of appropriate PRT, and effective symptom control at the point of referral and consultation. In a classic pre-consultation survey of patients with metastatic disease referred to PRT, 35% believed their cancer was curable, 20% expected PRT would cure it, and 38% believed PRT would prolong life, findings that underscore how misconceptions can shape decisions and goals of care [5]. In a dedicated PRT clinic study, misconceptions persisted even after consultation (17% before vs. 15% after believed PRT would cure their cancer), though expectations for symptom relief improved and treatment-related concerns declined, indicating that structured encounters can shift understanding and anxiety even as some beliefs endure [6]. In this context, structured patient education functions as a key enabler of balanced informed consent and shared decision-making in PRT, aligning expectations with palliative intent and eliciting patients’ values and goals. Importantly, misconceptions about radiation oncology are present even among medical students and primary care physicians; although higher training levels and decisive RO rotations are associated with better knowledge, substantial gaps persist, which may hinder timely referral and initiation of appropriate palliative radiotherapy [7]

In oncology more broadly, structured patient education—standardized, protocolized content delivered via counseling, print, or multimedia—has demonstrated benefits for knowledge acquisition, anxiety reduction, self-management, and shared decision-making, and is endorsed by professional guidance on patient–clinician communication [8,9]. Among adults referred to or receiving PRT, emerging studies likewise suggest beneficial effects across several endpoints, including knowledge, decisional readiness or uncertainty, and symptom outcomes such as pain and related clusters, using concise multimedia tools and nurse-delivered encounters [10,11,12,13].

Yet, compared with the broader oncology education literature, evidence for structured patient education within PRT pathways remains scarce and scattered across heterogeneous designs (e.g., clinic-based counseling, psychoeducational sessions timed to treatment, and multimedia decision supports). There is no consolidated synthesis describing what is taught, when and by whom it is delivered, and how these approaches influence clinical (pain/symptom), knowledge/decision, and psychosocial outcomes for adults referred to or receiving PRT. Addressing this gap is timely: better education could mitigate misconceptions, facilitate earlier and more appropriate referrals, align expectations with the true goals of PRT, and ultimately enhance patient-centered outcomes.

PRT is delivered across heterogeneous scenarios that vary by patient and tumors context (e.g., symptom profile, performance status, and estimated prognosis), care setting and resources (inpatient vs. outpatient; availability of palliative-care services; language and health literacy), and phase along the palliative trajectory, from first recognition of metastatic disease to the last weeks of life. These dimensions shape the goals and feasibility of education and communication, and influence when and how PRT is discussed (decision-focused at referral, self-management–focused during treatment, and comfort-oriented near end-of-life) [1,2,3,4,10,12,13].

Based on this background, this narrative review synthesizes current evidence on structured patient education and on studies that characterize education/communication content, informational needs, or decision processes among adults referred to or receiving PRT, describing intervention/content scope, timing, delivery personnel/modalities, and effects on clinical (e.g., pain and symptom relief, function, quality of life), knowledge-based (e.g., knowledge, decisional conflict/readiness, expectations), and psychosocial outcomes.

## 2. Materials and Methods

### 2.1. Review Design and Reporting Framework

This work is a narrative review conducted and reported in alignment with the six SANRA (Scale for the Assessment of Narrative Review Articles) quality domains: (i) justification of the article’s importance for readers; (ii) statement of the aims; (iii) description of the literature search; (iv) referencing; (v) scientific reasoning; and (vi) appropriate presentation of data [14]. Given the heterogeneity in study designs, interventions, and endpoints across the eligible literature, we did not plan or perform a meta-analysis, a formal risk-of-bias assessment, or a Preferred Reporting Items for Systematic Reviews and Meta-Analyses (PRISMA)-style systematic synthesis. A completed SANRA checklist is provided in Appendix B. For transparency, although this is a narrative review, we report the identification and screening yield using a PRISMA-2020 flow diagram and provide a table of full-text exclusions with reasons (see Appendix A).

### 2.2. Eligibility Criteria

We included peer-reviewed, English-language original studies (any design: randomized, non-randomized, pre–post, cohort, cross-sectional, and qualitative) published from 1 January 2000 to 18 July 2025 that evaluated structured patient-education interventions or characterized education/communication content, informational needs, or decision processes among adults (≥18 years) referred to or receiving PRT, irrespective of primary tumor type or anatomical site. “Structured patient education” was defined as standardized or pre-specified patient-facing content or processes (e.g., scripted nurse counseling, psychoeducation curricula, decision-aid videos, structured clinic consultations, printed/graphic materials with defined content, and programs with scheduled follow-ups). Studies enrolling mixed treatment populations were eligible only if (a) the cohort consisted of patients referred to PRT, or (b) results were reported separately for PRT recipients; otherwise they were excluded from data extraction. We excluded reviews, editorials, opinion papers, conference abstracts without full data, studies focused solely on clinician education without a patient component, curative-intent radiotherapy, pediatric populations, and general palliative-care education not linked to PRT pathways. We set 1 January 2000 as the lower bound to capture the modern era of PRT (short hypofractionated schedules, outpatient delivery) and contemporary patient-education modalities, maximizing applicability to current practice.

### 2.3. Information Sources and Search Strategy

We searched PubMed/MEDLINE, Scopus, and the Cochrane Library for records published from 1 January 2000 to 18 July 2025 (English language). Searches used combinations of controlled vocabulary and free-text terms pertaining to PRT, patient education, and education delivery formats. We supplemented database searches with backward and forward citation tracking and by screening reference lists of relevant reviews. Database-specific search strings are presented in Appendix A, and the number of records retrieved per database and at subsequent stages is summarized in the PRISMA-2020 flow diagram (Appendix A). The 2000 start year was chosen a priori to reflect contemporary PRT techniques and education delivery; earlier literature was considered less generalizable to current pathways.

### 2.4. Study Selection

All records were imported into a reference manager and de-duplicated prior to screening. Two authors (EG, CMD) independently screened titles/abstracts against eligibility criteria; potentially relevant records underwent full-text review by the same reviewers. Disagreements at either stage were resolved through discussion; persistent conflicts were adjudicated by the senior author (AGM). Reasons for exclusion at full text (e.g., curative-intent RT, education not structured, general palliative-care education without a PRT link, mixed cohorts without separable PRT data) were recorded. Counts at each screening stage and the list of full-text articles excluded with reasons are provided in Appendix A, respectively.

### 2.5. Data Extraction and Data Items

Using a piloted extraction template, EG and CMD independently extracted study characteristics and outcomes from each included paper. Extracted items comprised: study design, setting (inpatient/outpatient; consult/decision-aid context vs. during or after PRT), country/region, sample size, participant characteristics (tumor types, referral or treatment status), radiotherapy intent and regimen where available, education content (topic domains), delivery modality (in-person, telehealth/telephone, video, printed/graphic), personnel (e.g., nurse, radiation oncologist (RO)), timing (pre-consult/referral, during PRT, post-PRT), comparator (if any), follow-up duration, and outcomes. Outcomes were grouped a priori into (i) clinical (e.g., pain intensity/control; symptom clusters; functional status; quality of life), (ii) knowledge/decision (e.g., knowledge scores, decisional conflict/readiness, expectations), (iii) psychosocial (e.g., anxiety, satisfaction), and (iv) implementation/feasibility (e.g., adherence, acceptability). Any discrepancies were settled by consensus or by AGM.

### 2.6. Synthesis Approach

Given heterogeneity in interventions, comparators, and endpoints, we conducted a narrative synthesis. Studies were organized by when education occurred (pre-referral/consultation; during PRT; post-PRT) and by primary outcome domain(s). Where feasible, we report absolute and relative changes (e.g., proportions achieving pain control; pre–post knowledge gains) and highlight effect directions and consistency across studies. No quantitative pooling, sensitivity analyses, or formal quality/risk-of-bias assessments were undertaken due to methodological diversity and the narrative scope, consistent with SANRA guidance for narrative reviews [14].

### 2.7. Ethics

This review synthesizes previously published data and did not involve human subjects or access to identifiable patient information; therefore, institutional review board approval and informed consent were not required.

## 3. Results

Six studies met the eligibility criteria and were included in the narrative synthesis [6,10,12,13,15,16]. Consistent with our a priori window (2000–2025), the included reports spanned 2005–2023. The screening flow is depicted in Appendix A. A summary of the studies included is presented in Table 1. Moreover, Table 2 summarizes outcome signals across the included studies (+: improvement; –: worsening; /: no change; NA: not assessed). Geographically, studies originated from the Netherlands (two studies) [13,16], the United States (two) [10,15], Canada (one) [6], and Hong Kong, China (one) [12]. The designs comprised two randomized controlled trials (RCTs) (a nurse-led psychoeducational intervention during/around PRT in advanced lung cancer [12] and a multicenter nurse-led Pain Education Program adjunctive to PRT for painful bone metastases [13]); two prospective pre–post studies focused on patient knowledge/decision outcomes at the time of PRT referral/consultation [6,10]; one qualitative interview study exploring goals of care and prognostic understanding among patients during their first PRT course [15]; and one observational communication study using Roter Interaction Analysis System (RIAS) coding of videotaped initial PRT consultations [16]. Settings spanned inpatient (decision-aid videos among hospitalized patients [10]) and outpatient environments (dedicated PRT clinic [6], radiotherapy departments [12,13,16]).

### 3.1. Narrative Synthesis

Across the included studies, settings ranged from inpatient pre-decision hospitalizations to outpatient clinics, and populations spanned patients newly referred to PRT to those already on treatment, reflecting heterogeneity in indications and phases of care [6,10,12,13,15,16].

#### 3.1.1. Pre-Referral/Consultation Education and Decision Support

At the point of referral or initial consultation, structured education improved decision quality but revealed persistent misconceptions. In a dedicated PRT clinic, Mitera et al. observed increased expectations for symptom relief and lower anxiety and treatment concerns after consultation, yet beliefs that PRT or cancer was curable did not materially change [6]. Among hospitalized patients considering PRT, a brief decision-aid video significantly increased PRT knowledge, reduced decisional uncertainty, and raised readiness to proceed with PRT; acceptability was high, although readiness for palliative care consultation did not change [10]. Qualitative interviews during patients’ first PRT course highlighted that many use restorative (function/quality-of-life) language, whereas a subset default to combat/cure framing; participants preferred physicians as the primary source of prognostic information and requested written materials, reinforcing the role for concise, standardized educational tools aligned with RO counseling [15]. Communication analysis of initial PRT consultations showed limited patient medical question-asking, with decisions often pre-determined before the consultation with the RO, underscoring the need to proactively invite questions, clarify goals, and normalize palliative intent [16].

#### 3.1.2. Education Integrated with PRT Delivery

Two interventional trials embedded structured education within the PRT pathway. In advanced lung cancer, a nurse-delivered psychoeducational package (with relaxation coaching) administered shortly before and during PRT produced favorable changes in the dyspnea, fatigue, and anxiety cluster and improved functional ability versus usual care [12]. In a larger multicenter trial, adding a nurse-led Pain Education Program (standardized content plus scheduled phone reinforcement) to usual care in patients receiving PRT for painful bone metastases increased the proportion achieving pain control at 12 weeks and accelerated time-to-control, though global quality of life did not differ between arms [13]. These results suggest symptom-targeted education, particularly when reinforced over time, can produce clinically meaningful improvements complementary to the symptomatic effect of PRT itself.

Effective programs were concise (e.g., a brief video) [10] or structured yet personalized (nurse-led interviews with myth-busting, medication instruction, and pain-diary use, plus follow-up calls) [13]. Interventions were predominantly nurse-delivered [12,13], with ROs central to consultation-based education [6,16]. Common gaps across studies included persistent curative expectations despite consultation [6,15], potential under-addressing of palliative-care uptake/readiness [10], and unchanged overall quality of life despite improved pain metrics [13], all pointing to opportunities for multi-modal content (expectations setting, shared decision-making prompts, symptom self-management skills) and timing (pre-consultation priming plus post-treatment reinforcement).

## 4. Discussion

### 4.1. Principal Findings

Across six studies spanning 2005–2023 and four countries, we found limited but consistent literature describing patient education and communication within PRT pathways. Interventional studies suggest that structured education delivered alongside PRT can improve clinically relevant symptom outcomes, most notably faster and more frequent pain control when adjunctive pain education accompanies radiotherapy for painful bone metastases [13], and reductions in dyspnea, fatigue, and anxiety symptom clusters with modest functional gains in advanced lung cancer [12]. Pre-referral or consultation-time education primarily improved knowledge and decision quality, with a brief decision-aid video increasing knowledge and readiness and reducing decisional uncertainty among hospitalized candidates for PRT [10]. Observational and qualitative studies highlighted persistent curative misconceptions, the central role of physicians in prognostic communication, and limited patient question-asking during initial consultations [6,15,16]. In summary, as shown in Table 2, across all studies the observed effects are neutral or favorable, with no consistent signal of harm. The evidence base is highly heterogeneous in design, delivery, and outcome measures, with many endpoints not assessed (NA).

### 4.2. Interpretation and Implications for Practice

Targeted, standardized education can add value at two distinct points in the PRT pathway: before or at referral/consultation, to support informed decisions and calibrate expectations [6,10,15,16]; and during treatment, to reinforce symptom self-management and address analgesic use and beliefs, thereby complementing the radiobiological effects of PRT [12,13]. The recurring finding that global quality of life did not change despite improved pain metrics in the bone-metastasis trial [13] suggests that single-domain education (e.g., pain) may be insufficient to shift broader well-being and that multi-component approaches may be required. The observation that curative beliefs can persist after consultation [6,15] underscores the need for explicit goals-of-care language and take-home materials that reiterate palliative intent, consistent with communication guidance in oncology [9]. Brief formats (e.g., short videos) appear feasible in time-pressured settings [10], whereas nurse-delivered programs with scheduled follow-up may sustain behavior change beyond the initial visit [13].

Beyond patient-facing materials, adequate delivery of PRT depends on clinician-side knowledge and competencies. Core skills include prognostic clarity and goals-of-care communication, systematic symptom assessment and analgesia, and evidence-based selection of indications and fractionation. These competencies should be cultivated during radiation oncology residency and reinforced through interprofessional practice. For example, integrating a palliative-care consultation service with interdisciplinary ward rounds within radiation oncology has been reported to improve care coordination and deepen residents’ palliative-care knowledge, supporting high-quality, timely PRT and better-informed consent and shared decision-making [9,17].

Accordingly, education should be tailored to phase and context. At first referral, tools should clarify palliative intent, expected symptom benefits and time-to-effect, alternatives, and logistics; during PRT, coaching on symptom self-management and analgesia is paramount; and near end-of-life, discussions should explicitly address likely benefit versus burden and the option to defer or forgo PRT when not aligned with patient goals and available resources [1,2,3,4,10,12,13]

Particularly, when palliative RT is contemplated very near the end of life, education should explicitly weigh time-to-analgesia against time-remaining and set realistic expectations about what pain relief is likely and when. Patients and caregivers should be informed about the possibility of a transient pain flare and given a simple plan to prevent and manage it (e.g., short steroid courses when appropriate, clear rescue-analgesia instructions, and who to contact for uncontrolled pain). Shared decision-making should also cover the logistics and burden of treatment (immobilization, transport, number of visits), with a bias toward short, simple schedules when likely to meet the patient’s goals. At a service level, 30-day mortality after palliative RT is used as a quality indicator and can prompt reflective review of selection and timing [18]. In parallel, the increasing use of advanced techniques at the end of life has uncertain incremental value for near-term pain outcomes; these options should therefore be individualized and discussed in terms of potential benefit, burden, and opportunity costs within SDM [19].

Finally, in addition to symptom domains and expected analgesic benefit, patient education for PRT should make explicit the potential financial and time burdens of treatment, out-of-pocket costs, transport and caregiver time, and the opportunity costs of multiple visits, especially when life expectancy is limited. Framing these trade-offs helps patients and caregivers compare short, simple schedules (often fewer visits and lower logistical burden) with more complex/advanced techniques that may increase planning/immobilization time and costs without clear near-term gains in pain relief for many end-of-life scenarios. Such discussions should be embedded in shared decision-making and accompanied by a bridging analgesia plan if RT is deferred or shortened, consistent with the end-of-life RT literature that recommends incorporating financial and time toxicity into counseling [20]

### 4.3. Positioning Within the Wider Oncology Education Literature

Compared with other oncology settings, where patient education and decision support have repeatedly been associated with improvements in knowledge, anxiety, decision quality, adherence, and symptom control, the PRT literature remains sparse, with only two randomized trials [12,13] and few multi-site evaluations [6,10,12,13]. Across oncology, meta-analysis shows reductions in anxiety and pain and gains in knowledge with psychoeducational care [21]. Randomized and quasi-experimental studies indicate that education combined with monitoring or specialist input can enhance pain outcomes and daily functioning [22,23]. Systematic reviews report small-to-moderate improvements in cancer pain and adherence with structured education [24]. Patient decision aids consistently increase knowledge and produce more values-congruent choices [25]. A subsequent comprehensive review further supports the effectiveness of patient-based educational interventions for cancer-related pain [26]. Interventions around radiotherapy and chemotherapy workflows improve preparedness or reduce distress [27,28,29,30]. Syntheses extend these effects to survivors and caregivers [31], and a recent narrative overview of pain-education reviews reinforces these findings [32]. Together with upstream PRT-focused materials that improve awareness [11], these broader findings support the plausibility of education as a useful adjunct in PRT. The relative paucity of PRT-specific studies, despite the high prevalence of symptomatic metastatic disease and the established role of PRT in palliation [1,2,3], indicates an underdeveloped implementation evidence base.

In parallel, upstream clinician education remains essential: a multi-institutional U.S. survey documented persistent misconceptions about radiotherapy among medical students and primary care physicians, with improved—but still incomplete—knowledge among senior students and those completing RO rotations; such gaps can become barriers to timely PRT referral and initiation [7].

### 4.4. Limitations of the Included Evidence

The evidence is limited by small sample sizes (particularly in pre–post and qualitative work), single-center designs of most studies, and heterogeneity in content, timing, and delivery of education interventions. Many studies used pre–post designs without concurrent controls [6,10], which are vulnerable to secular trends and expectation effects. For the trials, blinding was not performed (as it is challenging for educational interventions), which may confer a risk of performance and detection bias [12,13]. Outcomes varied across studies (e.g., Decisional Conflict Scale subscales, Brief Pain Inventory, EORTC instruments), limiting cross-study comparability. Follow-up was short in most reports (weeks rather than months), constraining conclusions about durability. Generalizability was also limited: populations were predominantly from high-income academic settings, specific tumor sites (lung, bone) were over-represented, and data on language, literacy, and cultural tailoring were scarce. Notably, several studies documented unchanged overall quality of life despite improvements in targeted symptoms [13], and none assessed downstream outcomes such as referral timing, healthcare utilization, or concordance of care with patient goals.

Finally, by design we excluded studies focused solely on clinician education without a patient component (see Eligibility Criteria, §2.2), which is a limitation given that clinician competencies developed during residency and through interprofessional training are foundational to effective patient education and communication in PRT [17].

### 4.5. Strengths and Limitations of This Review

This review followed SANRA guidance for narrative reviews and used duplicate screening and extraction to enhance rigor. However, as a narrative synthesis without formal risk-of-bias assessment or quantitative pooling, our conclusions are interpretive rather than definitive. We restricted inclusion to English-language publications and searched three databases (PubMed/MEDLINE, Scopus, Cochrane Library), supplemented by citation chasing; relevant reports indexed elsewhere may have been missed. The small number of studies and their heterogeneity precluded assessment of publication bias. Finally, by broadening eligibility to include studies that characterized education/communication content and informational needs within PRT (and not only structured interventions), we increased conceptual coverage but also heterogeneity.

### 4.6. Directions for Future Research

Priority areas include pragmatic, multi-site trials of education bundles that combine pre-consultation priming (e.g., brief video or low-literacy handout), structured consultation tools (checklists, scripted language for palliative intent), and post-treatment reinforcement (nurse follow-ups). Trials should measure standardized symptoms and decision outcomes, report durability beyond 12 weeks, and incorporate patient-important endpoints such as goal-concordant care and caregiver outcomes. Equity-focused work is needed to evaluate language access, health literacy, and cultural tailoring, and to test delivery via telehealth/telephone for patients with limited mobility. Moreover, implementation studies within routine PRT services, using mixed-methods designs, could identify workflow-compatible strategies and sustainment factors. Furthermore, future PRT studies, particularly those centered on pain education, should prospectively measure patient-reported financial toxicity and time burden (e.g., visit counts, travel/time costs) and test whether cost/time discussions within education bundles improve decision quality, goal-concordant care, and patient-reported pain outcomes [20]. Finally, prospective studies should test pain-education bundles tailored to end-of-life PRT (time-to-benefit communication, pain-flare prophylaxis, bridging analgesia plans, early follow-up) and evaluate whether advanced techniques near the end of life meaningfully improve patient-reported pain relief, caregiver outcomes, and resource use, alongside quality indicators such as 30-day mortality [18,19].

## 5. Conclusions

While the evidence base is limited, available studies suggest that structured education and deliberate communication for patients undergoing PRT can improve targeted outcomes, pain control, symptom clusters, and decision quality, without clear harms. Given the central role of PRT in palliative oncology and the high prevalence of misconceptions and late referrals [[1],[2],[3],[4],[5],[6],[15],[16],[33],[34],[35],[36],,,[37],[38],[39],[40],[41],[42]], developing and evaluating scalable, standardized educational approaches appears warranted. The small number of heterogeneous studies underscores the need for methodologically robust research to guide routine implementation.

## Figures and Tables

**Table 1 cancers-17-03109-t001:** Summary of the studies included.

Study (Year)	Country/ Setting	Design and Population	Phase of Care (Relative to PRT)	Education/Communication Intervention	Primary Outcomes	Key Findings
Mitera et al. (2012) [6]	Canada; dedicated PRT clinic	Prospective pre–post survey (n = 100) of patients referred for PRT	Pre- and post-consultation with RO	Structured consultation; printed information as per clinic routine	Understanding/expectations; anxiety; satisfaction	Symptom-relief expectations improved; treatment concerns/anxiety decreased; belief in cure unchanged (17% → 15%); life-prolongation expectations unchanged.
Dharmarajan et al. (2019) [10]	USA; tertiary cancer center (inpatient)	Prospective single-arm pre–post (n = 40) among hospitalized patients referred to PRT	Prior to decision-making during hospitalization	Brief decision-aid video (PRT + palliative care)	Decisional uncertainty; knowledge; readiness	Knowledge increased (≈60 → 88% correct); uncertainty reduced; readiness for PRT increased; high acceptability; readiness for palliative care unchanged.
Chan et al. (2011) [12]	Hong Kong, China; public hospital	RCT; advanced lung cancer receiving PRT	1-week pre-PRT and week 3 of radiotherapy	Nurse-delivered psychoeducation + relaxation training	Symptom cluster (dyspnea, fatigue, and anxiety); function	Significant time × group effects: improvements in breathlessness, fatigue, anxiety, and functional ability vs. usual care.
Geerling et al. (2023) [13]	The Netherlands; five radiotherapy centers	Multicenter RCT (n = 308; 182 completed) in patients receiving PRT for painful bone metastases	Before radiotherapy with follow-up at 1, 4, 8, and 12 weeks	Nurse-led Pain Education Program + phone reinforcement	Pain control at 12 weeks; time-to-control; pain scores; quality of life	71% vs. 52% achieved pain control at 12 weeks (*p* = 0.008); faster time-to-control (29 vs. 56 days; *p* = 0.003); greater pain reduction; quality of life similar between groups.
Chen et al. (2022) [15]	USA; comprehensive cancer center	Qualitative interviews (n = 17) during first PRT for bone/lung metastases	During or within one month after PRT	Semi-structured interviews on goals/prognosis and informational needs	Qualitative themes on goals framing, prognosis understanding, and communication preferences (semi-structured interviews; thematic analysis).	Mixed “restorative vs. combat” goal language; some curative misconceptions; strong preference for prognostic info from physicians and desire for written materials.
Timmermans et al. (2005) [16]	The Netherlands; academic radiotherapy department	Observational video-recorded initial PRT consultations; RIAS analysis	Initial consultation prior to/at start of PRT	No formal education program; analysis of participation prompts	RIAS-coded communication in initial palliative RT consult: patient participation, RO facilitation, and presence/absence of decision discussion; no clinical/RO outcomes	Patient medical question-asking was low; ROs mainly provided information and invited narratives; treatment decisions were rarely discussed (often pre-decided).

Abbreviations: PRT, palliative radiotherapy; RCT, randomized controlled trial; RIAS, Roter Interaction Analysis System; RO, radiation oncologist.

**Table 2 cancers-17-03109-t002:** Summary of outcomes of studies on patient education in palliative radiotherapy.

Study	Educational Intervention	Pain (Intensity)	QoL	Anxiety	Dyspnea	Fatigue	Function	PRT Knowledge	Decision Uncertainty	Readiness for PRT	Readiness for PC Consultation	Understanding of Prognosis, Role of RT	Satisfaction	Participation in Consultation	Expectations of Cure/Life Prolongation
Timmermans 2005 [16]	RIAS analysis of initial consultations (observational; no training)	NA	NA	NA	NA	NA	NA	NA	NA	NA	NA	NA	NA	/	NA
Chan 2011 [12]	Psychoeducation + progressive muscle relaxation	NA	NA	+	+	+	+	NA	NA	NA	NA	NA	NA	NA	NA
Mitera 2012 [6]	Standard RT consultation; pre/post survey	NA	NA	+	NA	NA	NA	NA	NA	NA	NA	+	/	NA	/
Dharmarajan 2019 [10]	In-hospital decision-aid video on PRT (pre/post assessments)	NA	NA	NA	NA	NA	NA	+	+	+	/	NA	/	NA	NA
Chen 2022 [15]	Qualitative interviews on PRT goals/prognosis	NA	NA	NA	NA	NA	NA	NA	NA	NA	NA	/	NA	NA	NA
Geerling 2023 [13]	Nurse-led PEP before RT (45–60′) + four follow-up calls	+	/	NA	NA	NA	/	NA	NA	NA	NA	NA	NA	NA	NA

LEGEND: Symbols: + = significant improvement; – = worsening; / = no change (or no difference vs. control in RCTs); NA = not assessed. Abbreviations: PRT = palliative radiotherapy; RT = radiotherapy; PC = palliative care; QoL = quality of life; RIAS = Roter Interaction Analysis System (coding of patient–clinician communication); PEP = Pain Education Program (nurse-led education); PMR = progressive muscle relaxation; RCT = randomized controlled trial; RO = radiation oncology; pre-/post- = assessed immediately before/after the intervention or consult; ′ = minutes; wk(s) = weeks.

## Data Availability

No new data were created or analyzed in this study. Data sharing is not applicable to this article.

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
