# Peer review of "Patient Education and Communication in Palliative Radiotherapy: A Narrative Review"

_cancers, 2025, doi:10.3390/cancers17193109_

Round 1

Reviewer 1 Report

Comments and Suggestions for Authors

The review „Patient education and communication in palliative radiotherapy: a narrative review” outlines the importance of patient education for a well-balanced informed consent. It therefore adds insights into the value of shard-decision making in complicated treatment scenarios.

  • Although patient information is indeed an important aspect of informed counselation, it should be emphasized that the main condition for adequate palliative (radiation) therapy are knowledge and competences on the physician site. This should be implemented during residency (e.g. see Strahlenther Onkol. 2023 Mar;199(3):251-257. doi: 10.1007/s00066-022-01989-0).
  • In this regard, the analyzed studies revealed the heterogeneity of palliative treatment scenarios. It should be described more clearly that palliative treatment situations may differ due to the patient's and tumor situation, resources and phase of palliative care (from first diagnosis of metastasis to end-of-life).
  • Misconceptions about RO are unfortunately even present during medical school. (Int J Radiat Oncol Biol Phys. 2016 Feb 1;94(2):235-42). Higher years of studies and decisive RO rotations may help to improve (but not resolve) these misbeliefs which may serve as a barrier to initiate adequate palliative treatment (also on the physician site).
  • Initiation of RT in an advanced palliative phase (e.g. in the last month of life) necessitates a critical pro-con argumentation, as a low percentage of RT in patients in the terminal phase is increasingly seen as a quality indicator (J Med Imaging Radiat Oncol. 2020 Aug;64(4):570-579. doi: 10.1111/1754-9485.13073). However, the use of advanced techniques for these patients increases (J Oncol Pract. 2014 Jul;10(4):e269-76. doi: 10.1200/JOP.2013.001348.) the reason and implications of which are uncertain. I feel that this particular treatment situation should be discussed.
  • Although the symptomatic domains and treatment efficacy are major contributors to initiate (or not) palliative radiatiation treatment, other aspects like financial toxicity should also be considered (and discussed in the review (Semin Radiat Oncol. 2023 Apr;33(2):203-210. doi: 10.1016/j.semradonc.2022.11.002.)
  • The last column of Table 1 is nearly impossible to read as there is no clear separation between the different studies. This has to be improved.

Minor:

In the introduction, I would not state rather state that side effects should be minimized in regard of the treatment goal of symptomatic improvement and that toxicities have to be carefully balanced with the intended treatment goal.

Introduction line 72: please replace “non-trivial” with a more scientific expression like “high” or “relevant”

Table 2 should also be re-designed, e.g. with the text arrangement in the top line.

Author Response

Comment 1:

The review „Patient education and communication in palliative radiotherapy: a narrative review” outlines the importance of patient education for a well-balanced informed consent. It therefore adds insights into the value of shard-decision making in complicated treatment scenarios.

Response 1:

We sincerely thank the reviewer for this positive assessment. We fully agree that balanced informed consent and shared decision-making are central to high-quality palliative radiotherapy care. To make this emphasis more explicit for readers, we have added a concise statement in the Introduction linking structured education to shared decision-making, and we ensured consistent use of the term “shared decision-making” throughout the manuscript (already present among the keywords).

Changes in the manuscript (location + exact text):

  • Section: Introduction, Paragraph 3 (end of paragraph; after the sentence ending with ref. [6]).

Inserted sentence:
“In this context, structured patient education functions as a key enabler of balanced informed consent and shared decision-making (SDM) in PRT, aligning expectations with palliative intent and eliciting patients’ values and goals.”

Comment 2:

Although patient information is indeed an important aspect of informed consultation, it should be emphasized that the main condition for adequate palliative (radiation) therapy are knowledge and competences on the physician site. This should be implemented during residency (e.g. see Strahlenther Onkol. 2023 Mar;199(3):251-257. doi: 10.1007/s00066-022-01989-0).

Response 2:

We thank the reviewer for this important point and fully agree that high-quality palliative radiotherapy rests on clinician-side competencies: prognostic clarity, goals-of-care communication, symptom assessment and analgesia, and evidence-based RT decision-making developed during residency and reinforced in practice. While our review purposely focused on patient-facing education-communication, we now make this clinician-training prerequisite explicit in the Discussion and acknowledge it as a scope limitation. We also cite evidence that interprofessional models embedded in radiation oncology (e.g., palliative-care consultation services with joint ward rounds) strengthen resident competencies and can improve patient pathways, thereby supporting informed consent and shared decision-making in PRT. [We reference ASCO communication guidance already cited in the manuscript and add Oertel et al., 2023.]

Changes in the manuscript (location + exact text):

  • Section: 2. Interpretation and implications for practice: inserted a new paragraph at the end of the section (after the existing paragraph).

Inserted paragraph:
“Beyond patient-facing materials, adequate delivery of PRT depends on clinician-side knowledge and competencies. Core skills include prognostic clarity and goals-of-care communication, systematic symptom assessment and analgesia, and evidence-based selection of indications and fractionation. These competencies should be cultivated during radiation oncology residency and reinforced through interprofessional practice. For example, integrating a palliative-care consultation service with interdisciplinary ward rounds within radiation oncology has been reported to improve care coordination and deepen residents’ palliative-care knowledge, supporting high-quality, timely PRT and better-informed consent and shared decision-making [8,28].”

  • Section: 5. Strengths and limitations of this review: appended the following sentence to the end of the section.

Inserted sentence:
“Finally, by design we excluded studies focused solely on clinician education without a patient component (see Eligibility Criteria, §2.2), which is a limitation given that clinician competencies developed during residency and through interprofessional training are foundational to effective patient education and communication in PRT [28].”

  • Section: References: added the following citation as a new reference [28].

[28] Oertel, M.; Schmidt, R.; Steike, D.R.; Eich, H.T.; Lenz, P. Palliative care on the radiation oncology ward—improvements in clinical care through interdisciplinary ward rounds. Strahlenther Onkol. 2023;199(3):251–257. doi:10.1007/s00066-022-01989-0. Epub 2022 Aug 11. PMCID: PMC9938032.

Comment 3:

In this regard, the analyzed studies revealed the heterogeneity of palliative treatment scenarios. It should be described more clearly that palliative treatment situations may differ due to the patient's and tumor situation, resources and phase of palliative care (from first diagnosis of metastasis to end-of-life).

Response 3:

We sincerely thank the reviewer for this valuable suggestion. We agree that palliative radiotherapy (PRT) is delivered across highly heterogeneous scenarios shaped by patient and tumor factors, available resources (inpatient vs. outpatient; access to palliative-care services; language/health literacy), and phase along the palliative trajectory, from first recognition of metastatic disease to the last weeks of life. To make this explicit, we have added (i) a clarifying paragraph at the end of the Introduction outlining these dimensions; (ii) a brief orienting sentence at the start of the Narrative synthesis to foreground heterogeneity across the included studies; and (iii) a practical tailoring statement in the Interpretation and implications for practice. Together, these changes highlight why content and timing of education should differ at referral, during treatment, and near end-of-life, with supporting citations to studies already included in our review.

Changes in the manuscript (location + exact text):

  • Section: Introduction: inserted as a new final paragraph (immediately after the paragraph beginning “Based on this background, this narrative review synthesizes…”).

Inserted paragraph:

“PRT is delivered across heterogeneous scenarios that vary by patient and tumour context (e.g., symptom profile, performance status, and estimated prognosis), care setting and resources (inpatient vs. outpatient; availability of palliative-care services; language and health literacy), and phase along the palliative trajectory, from first recognition of metastatic disease to the last weeks of life. These dimensions shape the goals and feasibility of education and communication, and influence when and how PRT is discussed (decision-focused at referral, self-management–focused during treatment, and comfort-oriented near end-of-life) [1–4,9,11,12].”

  • Section: 2. Narrative synthesis: inserted one orienting sentence as the first paragraph under the 3.2 header (before §3.2.1).

Inserted sentence:

“Across the included studies, settings ranged from inpatient pre-decision hospitalizations to outpatient clinics, and populations spanned patients newly referred for PRT to those already on treatment, reflecting heterogeneity in indications and phases of care [6,9,11,12,15].”

  • Section: 2. Interpretation and implications for practice: appended the following sentences at the end of the section.

Inserted sentences:

“Accordingly, education should be tailored to phase and context. At first referral, tools should clarify palliative intent, expected symptom benefits and time-to-effect, alternatives, and logistics; during PRT, coaching on symptom self-management and analgesia is paramount; and near end-of-life, discussions should explicitly address likely benefit versus burden and the option to defer or forgo PRT when not aligned with patient goals and available resources [1–4,9,11,12].”

Comment 4:

Misconceptions about RO are unfortunately even present during medical school. (Int J Radiat Oncol Biol Phys. 2016 Feb 1;94(2):235-42). Higher years of studies and decisive RO rotations may help to improve (but not resolve) these misbeliefs which may serve as a barrier to initiate adequate palliative treatment (also on the physician site).

Response 4:

We are grateful for this insightful comment and fully agree. Persistent misconceptions about radiation oncology among medical students and primary care physicians can delay referral and the initiation of appropriate palliative radiotherapy. The cited multi-institutional survey by Zaorsky et al. shows that although knowledge improves with seniority and dedicated RO rotations, important gaps remain, underscoring that clinician-side education should complement patient-facing education to support timely, high-quality PRT decisions.

Changes in the manuscript (location + exact text):

  • Section: Introduction, Paragraph 3: appended at the end of the paragraph (after the sentence ending with ref. [6]).

Inserted sentence:

“Importantly, misconceptions about radiation oncology are present even among medical students and primary care physicians; although higher training level and decisive RO rotations are associated with better knowledge, substantial gaps persist, which may hinder timely referral and initiation of appropriate palliative radiotherapy [Zaorsky 2016].”

  • Section: 3. Positioning within the wider oncology education literature, first paragraph, appended the following sentence at the end.

Inserted sentence:

“In parallel, upstream clinician education remains essential: a multi-institutional U.S. survey documented persistent misconceptions about radiotherapy among medical students and primary care physicians, with improved, but still incomplete, knowledge among senior students and those completing RO rotations; such gaps can become barriers to timely PRT referral and initiation [29]”

  • Section: References: added as new reference:

Zaorsky, N.G.; Shaikh, T.; Handorf, E.; Eastwick, G.; Hesney, A.; Scher, E.D.; Jones, R.T.; Showalter, T.N.; Avkshtol, V.; Rice, S.R.; Horwitz, E.M.; Meyer, J.E. What Are Medical Students in the United States Learning About Radiation Oncology? Results of a Multi-Institutional Survey. Int. J. Radiat. Oncol. Biol. Phys. 2016;94(2):235–242. doi:10.1016/j.ijrobp.2015.10.008. (Epub 2015 Oct 9)

Comment 5:

Initiation of RT in an advanced palliative phase (e.g. in the last month of life) necessitates a critical pro-con argumentation, as a low percentage of RT in patients in the terminal phase is increasingly seen as a quality indicator (J Med Imaging Radiat Oncol. 2020 Aug;64(4):570-579. doi: 10.1111/1754-9485.13073). However, the use of advanced techniques for these patients increases (J Oncol Pract. 2014 Jul;10(4):e269-76. doi: 10.1200/JOP.2013.001348.) the reason and implications of which are uncertain. I feel that this particular treatment situation should be discussed.

Response 5:

Thank you for this excellent point. We fully agree and we have reframed our discussion around pain education when PRT is considered very near end-of-life. In that time window, education should make explicit: (i) expected time-to-analgesia from PRT versus time-remaining; (ii) the possibility of a transient pain flare and how to prevent/manage it; (iii) the rationale for short, simple schedules (when appropriate) versus more complex/advanced techniques whose added treatment burden may not translate into near-term pain relief; and (iv) a clear bridging analgesia plan (rescue dosing, who to call, and early reassessment). We also link this to service-level quality indicators (e.g., 30-day mortality after palliative RT) and note the observed increase in advanced technologies at end of life, highlighting the need to individualize choices within shared decision-making and to study patient-reported pain outcomes in this setting (citing the articles you provided).

Changes in the manuscript (location + exact text):

  • Section: 2. Interpretation and implications for practice — replace the previously added paragraph on end-of-life PRT with the following, explicitly centered on pain education.

Added paragraph:

“ When PRT is contemplated very near end-of-life, education should explicitly weigh time-to-analgesia against time-remaining and set realistic expectations about what pain relief is likely and when. Patients and caregivers should be informed about the possibility of a transient pain flare and given a simple plan to prevent and manage it (e.g., short steroid courses when appropriate, clear rescue-analgesia instructions, who to contact for uncontrolled pain). Shared decision-making should also cover the logistics and burden of treatment (immobilization, transport, number of visits), with a bias toward short, simple schedules when likely to meet the patient’s goals. At a service level, 30-day mortality after PRT is used as a quality indicator and can prompt reflective review of selection and timing [30]. In parallel, the increasing use of advanced techniques at the end of life has uncertain incremental value for near-term pain outcomes; these options should therefore be individualized and discussed in terms of potential benefit, burden, and opportunity costs within shared decision making [31].”

  • Section: 6. Directions for future research: appended the following sentence at the end of the section:

“Prospective studies should test pain-education bundles tailored to end-of-life PRT (time-to-benefit communication, pain-flare prophylaxis, bridging analgesia plans, early follow-up) and evaluate whether advanced techniques near end of life meaningfully improve patient-reported pain relief, caregiver outcomes, and resource use, alongside quality indicators such as 30-day mortality [30,31].”

  • Section: References: added as new entries.

Kain, M.; Bennett, H.; Yi, M.; Robinson, B.; James, M. 30-day mortality following palliative radiotherapy. J. Med. Imaging Radiat. Oncol. 2020;64(4):570–579. doi:10.1111/1754-9485.13073.

Guadagnolo, B.A.; Liao, K.P.; Giordano, S.H.; Elting, L.S.; Buchholz, T.A.; Shih, Y.C. Increasing use of advanced radiation therapy technologies in the last 30 days of life among patients dying as a result of cancer in the United States. J. Oncol. Pract. 2014;10(4):e269–e276. doi:10.1200/JOP.2013.001348

Comment 6:

Although the symptomatic domains and treatment efficacy are major contributors to initiate (or not) palliative radiation treatment, other aspects like financial toxicity should also be considered (and discussed in the review (Semin Radiat Oncol. 2023 Apr;33(2):203-210. doi: 10.1016/j.semradonc.2022.11.002.)

Response 6:

We thank the reviewer for this thoughtful suggestion. While our review centers on patient-facing education and communication, with a special focus on pain education in PRT, we agree that financial toxicity and “time toxicity” (e.g., number of visits, transport, caregiver time, planning/immobilization) are integral to shared decision-making and should be explicitly addressed during counseling. To keep our scope coherent yet responsive, we have added a concise subsection in the Discussion that frames cost/time burden as part of pain-focused education (e.g., when comparing short, simple schedules with more complex techniques near end of life), and we point readers to the end-of-life RT literature highlighting these considerations. We also flag the need for future PRT-specific studies to measure patient-reported financial/time burden alongside pain outcomes and decision quality.

Changes in the manuscript (location + exact text):

  • Section: 2. Interpretation and implications for practice: inserted a new paragraph at the end of the section.

Inserted paragraph:

“In addition to symptom domains and expected analgesic benefit, patient education for PRT should make explicit the potential financial and time burdens of treatment, out-of-pocket costs, transport and caregiver time, and the opportunity costs of multiple visits, especially when life expectancy is limited. Framing these trade-offs helps patients and caregivers compare short, simple schedules (often fewer visits and lower logistical burden) with more complex/advanced techniques that may increase planning/immobilization time and costs without clear near-term gains in pain relief for many end-of-life scenarios. Such discussions should be embedded in shared decision-making and accompanied by a bridging analgesia plan if RT is deferred or shortened, consistent with end-of-life RT literature that recommends incorporating financial and time toxicity into counseling [].”

  • Section: 6. Directions for future research: appended the following sentence to the end of the section.

Inserted sentence:

“Future PRT studies, particularly those centered on pain education—should prospectively measure patient-reported financial toxicity and time burden (e.g., visit counts, travel/time costs) and test whether cost/time discussions within education bundles improve decision quality, goal-concordant care, and patient-reported pain outcomes [Yeramilli 2023].”

  • Section: References: added a new entry:

Yerramilli, D.; Johnstone, C.A. Radiation Therapy at the End of Life: Quality of Life and Financial Toxicity Considerations. Semin. Radiat. Oncol. 2023;33(2):203–210.

Comment 7:

The last column of Table 1 is nearly impossible to read as there is no clear separation between the different studies. This has to be improved.

Response 7:

Thanks for the advice. We've reformatted the table and hope it's clearer and more readable now.

Comment 8:

In the introduction, I would not state rather state that side effects should be minimized in regard of the treatment goal of symptomatic improvement and that toxicities have to be carefully balanced with the intended treatment goal.

Response 8:

Thank you for this clarifying suggestion. We agree that, in palliative radiotherapy (PRT), the symptomatic goal must explicitly guide counseling and plan selection, with side effects minimized and expected toxicities weighed against the anticipated palliative benefit. We have updated the Introduction to state this principle clearly, aligning with the paper’s focus on patient education and expectation setting.

Changes in the manuscript (location + exact text):

  • Section: Introduction, Paragraph 1; appended at the end of the paragraph.

Inserted sentence:
“Because the aim of PRT is symptomatic improvement, side effects should be minimized, and any potential toxicities must be carefully balanced against the expected palliative benefit and the patient’s goals of care [1,3].”

Comment 9:

Introduction line 72: please replace “non-trivial” with a more scientific expression like “high” or “relevant”

Response 9:

Thank you for your comment. We changed the text as suggested

Comment 10:

Table 2 should also be re-designed, e.g. with the text arrangement in the top line.

Response 10:

Thanks for the advice. We've reformatted the table, reducing the font size and adding borders between the cells; we hope it's now clearer and more readable.

Reviewer 2 Report

Comments and Suggestions for Authors

Thank you for the opportunity to review this interesting and well written review. There are just two points which should be addressed to improve the manuscript.

The authors should include details of the number of studies identified in the initial search and how many were excluded at each stage and reason for inclusion. A flowchart of the screening process should also be included.

Figure 1 should be included in the results section rather than the discussion.

Author Response

Comment 1:

Thank you for the opportunity to review this interesting and well written review. There are just two points which should be addressed to improve the manuscript. The authors should include details of the number of studies identified in the initial search and how many were excluded at each stage and reason for inclusion. A flowchart of the screening process should also be included.

Response 2:

We are grateful for the constructive suggestion. Although this is a narrative review (planned and reported per SANRA), we agree that adding transparent screening details will improve clarity. We have therefore provided, in the Supplementary Materials: (i) the database-specific search strategies for PubMed/MEDLINE, Scopus, and Cochrane; (ii) a PRISMA-2020 flow diagram summarizing the number of records at each stage; and (iii) a table of full-text articles excluded with reasons. We have also added brief cross-references in the Materials and Methods and Results sections to guide readers to these items.

Changes in the manuscript (location + exact text):

  • Section: 1. Review design and reporting framework: appended at the end of the paragraph.

Inserted sentence:

“For transparency, although this is a narrative review, we report the identification and screening yield using a PRISMA-2020 flow diagram and provide a table of full-text exclusions with reasons (see Supplementary Figure S1 and Supplementary Table S1).”

  • Section: 3. Information sources and search strategy: appended at the end of the paragraph.

Inserted sentence:
“Database-specific search strings are presented in Supplementary Methods S1, and the number of records retrieved per database and at subsequent stages is summarized in the PRISMA-2020 flow diagram (Supplementary Figure S1).”

  • Section: 4. Study selection: appended at the end of the paragraph.

Inserted sentence:

“Counts at each screening stage and the list of full-text articles excluded with reasons are provided in Supplementary Figure S1 and Supplementary Table S1, respectively.”

  • Section: Results: append after the first sentence (“Six studies met the eligibility criteria…”).

Inserted sentence:

“The screening flow is depicted in Supplementary Figure S2.”

  • Section: Supplementary Materials: replace the existing sentence with the following.

Replacement text:

“Supplementary Materials: S1, Database-Specific Search Strategies (search close: 18 July 2025); Supplementary Figure 1, PRISMA-2020 Flow Diagram of study selection; Supplementary Table 1, Full-text records excluded with reasons.”

Comment 2:

Figure 1 should be included in the results section rather than the discussion.

Response 2:

Thank you, we agree. We have moved the visual summary from the Discussion into the Results and, per journal style, converted Figure 1 into Table 2. We also added an explicit cross-reference to Table 2 within the Results text and updated the mention in the Discussion accordingly.

Changes in the manuscript (location + exact text):

  • Section: Results: after the sentence “A summary of included studies is presented in Table 1.” added:

Inserted sentence:
“Table 2 summarizes outcome signals across the included studies (+: improvement; –: worsening; /: no change; NA: not assessed).”

  1. Section: 4. Discussion → 4.1. Principal findings: replaced the clause referencing the figure.

Original: “In summary, as illustrated in Figure 1, across all studies the observed effects are neutral or favorable, with no consistent signal of harm.”
Replacement: “In summary, as shown in Table 2, across all studies the observed effects are neutral or favorable, with no consistent signal of harm.”

Reviewer 3 Report

Comments and Suggestions for Authors

This narrative review is interesting, but I have one question about it. Why authors restricted date from 2020?

Author Response

Comment 1:

This narrative review is interesting, but I have one question about it. Why authors restricted date from 2020?

Response 1:

Thank you for raising this; our intent was not to restrict the evidence to 2020 onward. As specified in the Eligibility criteria and Information sources, our window spans 1 January 2000 to 18 July 2025; the six included studies indeed range from 2005 to 2023. We have clarified this explicitly in the Methods to avoid any ambiguity and added one sentence explaining why 2000 was chosen (to focus on contemporary PRT practice and modern education modalities).

Changes in the manuscript (location + exact text):

  • Section: 2. Eligibility criteria: appended at the end of the paragraph.

Inserted sentence:
“We set 1 January 2000 as the lower bound to capture the modern era of palliative radiotherapy (short hypofractionated schedules, outpatient delivery) and contemporary patient-education modalities, maximizing applicability to current practice.”

  • Section: 3. Information sources and search strategy: keep the stated range and added a brief rationale at the end of the paragraph.

Inserted sentence:
“The 2000 start year was chosen a priori to reflect contemporary PRT techniques and education delivery; earlier literature was considered less generalizable to current pathways.”

  • Section: Results after the first sentence.

Inserted sentence:
“Consistent with our a priori window (2000–2025), the included reports span 2005–2023.” 

Round 2

Reviewer 1 Report

Comments and Suggestions for Authors

Thank you for your rigorous review and implementation of my comments. I consent to publication!